

# A multi-queue-based ECN marking strategy for multi-class QoS guarantee in programmable networks

Yazhi Liu[1], Xinyi Yao[1], Zhigang Yang[2] and Wei Li[1]

[1] College of Artificial Intelligence, North China University of Science and Technology, Tangshan, China
[2] College of Electrical Engineering, North China University of Science and Technology, Tangshan, China

## ABSTRACT

Currently, network applications are experiencing explosive growth, and various types of network applications are showing a trend of varied demands for quality of network service. However, the existing Explicit Congestion Notification (ECN) marking methods have not taken into account the diversified Quality of Service (QoS) requirements of network applications. This article introduces a multi-queue ECN marking strategy targeting multiple QoS guarantees. The strategy utilizes virtual queues and dynamic weighted round-robin scheduling to achieve traffic partitioning in a programmable data plane. It constructs a multi-queue, multi-class QoS queuing model based on the QoS requirements of different traffic and network conditions. The model is solved by real-time to obtain the ECN marking thresholds and round-robin weights for different queues, in order to achieve dynamic QoS requirements of different network applications. We implemented this strategy in Mininet and BMv2, and compared it with DCQCN, P4QCN, and TCN. The experimental results indicate that this policy demonstrates good performance in terms of queue length, RTT, and throughput, while also ensuring fairness between traffics. Results of the experiment indicate that the proposed approach is superior to DCQCN and P4QCN in the field of performance fluctuation and rapid feedback, and it exhibits notable advantages over TCN, and also ensures the fairness of traffic.

Corresponding author
Zhigang Yang, yzg@ncst.edu.cn, yzg-lzy@163.com

# INTRODUCTION

With the rapid of cloud applications and services, the variety of traffic within data centers has significantly increased (*Gao et al., 2021*). Consequently, network congestion occurs frequently, leading to increased packet loss, higher latency, and decreased throughput in datacenter networks. Ensuring Quality of Service (QoS) for various types of traffic has become a new challenge for the next generation of networks (*Chen et al., 2019*). To address these network challenges, Explicit Congestion Notification (ECN) has been deployed as an effective tool within networks (*Pan et al., 2018*). Congestion control protocols based on ECN achieved favorable results within networks (*Kundel et al., 2021*). However, if the ECN threshold is configured improperly, it can have a significant impact on the performance of

the network. Therefore, efficient configuration of the ECN marking threshold is a crucial part of the process.

Currently, research on ECN marking typically encompasses the following two scenarios:

- **Assuming there is only one queue at the switch port.** In this scenario, the ECN strategy will mark all packets at the port based on port information (such as the number of packets or delay at the port), as seen in Data Center Transmission Control Protocol (DCTCP) (*Alizadeh et al., 2010*). However, the current trend in the data center industry is that each port of the switch has multiple queues (*Kim & Lee, 2021*). Therefore, Multi-Queue Explicit Congestion Notification (MQ-ECN) (*Bai et al., 2016b*) has been proposed to set ECN thresholds for each individual queue at the port. However, MQ-ECN still adjusts the ECN marking threshold for each queue based on the queue weight ratio, which fails to guarantee QoS requirements for each type of traffic. Furthermore, MQ-ECN is limited to weighted round-robin scheduling algorithms. Unlike forwarding, scheduling behavior is primarily determined through hardware configuration (*Sivaraman et al., 2016*). This results in poor portability and operability of MQ-ECN.

- **The application of ECN marking is limited to the TCP protocol.** Since the ECN marking mechanism relies on the TCP protocol, most research is typically conducted in TCP environments. However, as the demand for network performance increases, the proportion of UDP packets in the network is also gradually rising. Currently, research on transport mechanisms based on UDP, such as QUIC (*Langley et al., 2017*), is also rapidly advancing. In fact, we have observed that both UDP streams and TCP streams exhibit similar performance issues under certain traffic patterns.

Indeed, while the specific details may vary, most congestion control mechanisms in modern data center networks combine the use of port congestion control, First-In First-Out (FIFO) queues at switches, or the congestion feedback information (such as delay or switch state) from the end-to-end loop to achieve lower congestion.

Based on the background mentioned, this article introduces a multi-queue ECN marking strategy based on virtual queue. This article investigates virtual queue partitioning on programmable data planes to achieve multi-queue ECN marking, aiming to guarantee QoS requirements for various traffic types. The ECN marking is implemented in the data plane, only marking the last two bits of the TOS field in the IP packet header of ECN feedback packets, making it applicable to all IP packets. This strategy is implemented based on a programmable data plane, decoupling it from protocols and devices, making it easy to deploy. We have implemented this strategy on a software platform, and it has ultimately achieved the expected results.

The contributions of this article can be summarized as follows:

- We establish a multi-queue QoS objective optimization model based on queue theory, by considering that traffic has different QoS requirements and queues need different ECN markings. We take into account the relationship between marking thresholds, queue weights, and QoS objectives, and utilize optimization theory to solve the model, in order to obtain the optimal queue marking thresholds, queue weights, and optimal

arrival rate of packets. With this approach, we can effectively manage congestion and guarantee that each traffic flow receives the required QoS metrics according to its specific needs.

- We propose a dynamically weighted queue scheduling method based on virtual queues. This method, upon the arrival of traffic at the switch, divides it into distinct virtual queues based on packet header information. By employing a multi-queue QoS optimization model, the method dynamically determines queue weights, regulating the dequeue sequence of packets to achieve equitable scheduling among different traffic streams.

- We simulated the DCQCN (*Zhu et al., 2015*), P4QCN (*Geng, Yan & Zhang, 2019*), and TCN (*Bai et al., 2016a*) mechanisms (the code has been uploaded to GitHub (https://github.com/YXY-1998/CC)) on a programmable data plane software platform composed of Mininet and BMv2, and compared them with our proposed solution. The experimental results indicate that our solution effectively reduces queue length and transmission latency within the switches while ensuring throughput and maintaining stability in throughput variations. It has achieved the desired results as expected.

The remaining part of the article proceeds as follows: Section "Related Work" lists the current research results on congestion control methods and programmable queue scheduling. We identify the limitations of existing ECN strategies in Section "Motivation". Section "Design" is connected with the design of our ECN marking strategy. Section "Experiment" provides detailed information on the experimental setup and results, and in the end, Section "Conclusion and Future Work" summarizes the work of this article and points out the next research direction.

# RELATED WORK

In this section, we primarily discuss two aspects of related work: congestion control strategies based on ECN marking and programmable queue scheduling.

## Congestion control strategies

The research on congestion control methods based on ECN marking has been continuously improved with the development of data center networks. Here, we provide a brief overview of the evolution of ECN mechanisms from three perspectives.

## Traditional ECN strategies

D2TCP (*Vamanan, Hasan & Vijaykumar, 2012*) and L2DCT (*Munir et al., 2013*) have modified the DCTCP congestion control algorithm to meet the time constraints of network communication. The CEDM (*Shan & Ren, 2017*), proposed by *Shan & Ren (2017)*, accurately marks ECN to reduce throughput loss by minimizing queue oscillations. ECN (*Wu et al., 2012*) utilizes the standard TCP congestion control algorithm on the terminal host and employs ECN marking based on the instantaneous queue length. These solutions employ the same Active Queue Management (AQM) marking as DCTCP. Consequently, they are disadvantaged by increased queuing delay in the presence of RTT variations. DCQCN (*Zhu et al., 2015*) is an improved approach combining DCTCP and

QCN, which is a rate-based congestion control method enabling Remote Direct Memory Access(RDMA) in data centers.

## Multi-Queue ECN marking strategies

MQ-ECN (*Bai et al., 2016b*) first identified the limitations of existing ECN/RED implementations in packet scheduling and proposed a method to automatically adjust ECN thresholds based on network load. To overcome the limitations of MQ-ECN on packet schedulers, TCN (*Bai et al., 2016a*) proposes using instantaneous dwell time to mark packets. *Pan et al. (2018)* proposed a more accurate congestion detection mechanism by adding RTT thresholds at the port level based on the ECN marking strategy.

## ECN strategies combined with various methods

ECN# (*Zhang, Bai & Chen, 2019*) demonstrates significant variations in actual RTT in production environments (approximately three-fold), making it challenging to find appropriate static thresholds to balance queue occupancy and throughput. In the context of artificial intelligence, network technologies are also moving towards intelligent directions, combining ECN with various techniques. ECN-CoDel (*Alwahab & Laki, 2020*) implements the active queue management algorithm, CoDel (*Nichols et al., 2018*), using a packet processor (P4) independent of the programming protocol. When the dwell time of a packet exceeds the target time, it is no longer forwarded and waits for the first-round interval (100 ms) to enter the first drop state. All forwarded packets are marked with ECN to reduce the load on router queues. ECN-CoDel can significantly reduce retransmissions and achieve an almost zero packet loss rate. P4QCN (*Geng, Yan & Zhang, 2019*) is an enhanced Quantized Congestion Notification (QCN) design based on P4, implementing the QCN protocol in an IP network. However, it still adopts a static ECN marking strategy. EECN (*Shahzad et al., 2020*) provides an enhanced ECN mechanism where, upon congestion occurrence, switches intercept acknowledgement (ACK) data segments from receivers within the network and set the ECN-Echo bit using P4 programming. This achieves fast congestion notification without requiring additional control traffic. *Jiang & Zhang (2019)* propose an accurate congestion control mechanism that uses In-band Network Telemetry (INT) to capture network capacity from a global perspective and calculates the expected bandwidth for each traffic flow, utilizing the Adjusted Advertised Window (AAW) method to accurately feedback the actual capacity of the network to source nodes. Using machine learning, the active ECN dynamic marking strategy adaptive cruise control (ACC) (*Yan et al., 2021*) is introduced. ACC works in a distributed manner and dynamically computes marking thresholds using reinforcement learning to adapt to dynamic traffic patterns. QoSTCP (*Chen, Fang & Iqbal, 2020*) uses trTCM meters to determine if the traffic exceeds a certain threshold. When the traffic exceeds a specific rate, the ECN bit is marked as Rate Limiting Notification (RLN), and the congestion window growth is proportional to the rate of packets marked as RLN. *Laraba et al. (2020)* model ECN as an Extended Finite State Machine (EFSM) and store states and variables in registers. If the terminal host does not comply with the specified ECN state machine, packets may be dropped, or incorrect behaviors are corrected whenever possible. ECLAT (*Kim, Im &*

*Lee, 2021*) introduces a delay budget to adaptively control congestion markings, analyzing the cwnd (congestion window) growth pattern of TCP to estimate the time it takes for the sender's cwnd to reach the target cwnd and perform ECN markings based on the calculated time. PACC (*Zhong et al., 2022*) accurately calculates the real-time queue length based on a PI controller and timely returns Congestion Notification Packets(CNPs) to specific sources. SB-CC (*Wei et al., 2020*) utilizes the ECN mechanism to estimate the congestion level of each subflow and controls the congestion window size of each subflow based on different congestion levels to ensure fairness in subflow network loads.

The reference (*Brouwer & de Jager, 2023*) leverages programmable switches to provide high-speed processing capabilities for running AQM algorithms, analyzing implementation details of various AQM algorithms within the programmable data plane. This analysis assists researchers in better understanding and selecting AQM algorithms. iRED (*de Almeida et al., 2022*) exploits the programmability of the data plane by discarding packets at the ingress pipeline instead of at the egress, improving QoS for adaptive video streams compared to other AQM strategies based on drop decisions at the egress. BRT (*Zhang et al., 2024*) employs traffic-characteristic windows to detect if queues are in a persistent long-queue state. It adjusts the total buffer allocation for each traffic type based on the number of persistently long queues, reducing unnecessary buffering for meaningless queuing. Lastly, it computes buffer thresholds for RDMA/TCP queues separately and uses a simple yet effective method to prioritize the absorption of small flows. BRT effectively optimizes the networking performance of RDMA/TCP hybrid flows.

## Programmable queue scheduling

By using a programmable scheduler, the scheduling algorithm can be seen as a program running on a programmable switch chip, eliminating the need for hardware redesign when modifications to the algorithm are required. *Sivaraman et al. (2016)* proposed a programmable scheduler that can utilize a Push-In First-Out (PIFO) priority queue to schedule packets based on priority or temporal order without requiring changes to the switch chip. PR-AQM (*Li et al., 2023*) approximates PIFO behavior using limited FIFO queues. PR-AQM dynamically adjusts the mapping between packet ordering and queues based on the latency states of different priority queues, thereby achieving packet prioritization. *Lhamo et al. (2024)* points out that existing AQM mechanisms running on P4 typically do not support QoS. Therefore, it combines the well-known CoDel mechanism with static priority scheduling to achieve QoS differentiation in the data plane implementation of AQM. In order to minimize scheduling errors relative to the assumed PIFO implementation, SP-PIFO (*Alcoz, Dietmüller & Vanbever, 2020*) dynamically adjusts the mapping between priorities and SP queues. Building upon SP-PIFO, *Vass, Sarkadi & Rétvári (2022)* introduced an optimal offline scheme that outputs the best SP-PIFO configuration in polynomial time under a given random input model. *Yu et al. (2021)* proposed an Admission-In First-Out (AIFO) queue, which utilizes a single FIFO queue and determines packet admission based on the relative ranking of the packets. This approach enables packet scheduling and reduces hardware resource consumption.

In summary, this article designs a multi-queue ECN marking strategy aimed at QoS assurance. Leveraging the programmability of the data plane, the strategy's applicability is enhanced, as it supports all IP protocol packets compared to previous ECN marking policies. Additionally, inspired by mechanisms from SP-PIFO for dynamically adjusting packet priorities and mappings between SP queues, the article employs a virtual queue-to-priority queue mapping to simulate round-robin scheduling, ensuring fairness among flows. Building upon the M/M/1/N queuing model, this work constructs a multi-traffic objective optimization model. It dynamically computes queue scheduling weights and ECN marking thresholds based on QoS objectives and current queue states, considering diverse traffic demands to achieve equilibrium in competition. The optimal solutions for queue weights and ECN marking thresholds are derived through model solving techniques.

## MOTIVATION

In recent times, research centers and data center operators have proposed various algorithms based on ECN marking with the aim of optimizing the performance of data centers, including DCTCP (*Alizadeh et al., 2010*), DCQCN (*Zhu et al., 2015*), MQ-ECN (*Bai et al., 2016b*), TCN (*Bai et al., 2016a*), and ECN# (*Zhang, Bai & Chen, 2019*), among others. The primary objectives of these algorithms are to achieve high throughput and reduce tail latency. However, during our investigation and research into congestion control algorithms, we found certain limitations in the existing ECN algorithms.

- **Dependent on protocols, poor universality.** Some algorithms, improved based on DCTCP, notify the sender about congestion in the link by marking the ECE bit of TCP packets at the receiving end. At the same time, these congestion control strategies assume that all nodes in the network use ECN-based transport protocols (such as TCP, RDMA). These protocols use their specific packet headers, and if other headers are used, these strategies will not function properly. (RDMA can mark the IP header as an ECN indicator, but end-to-end control still requires the generation of specific format Congestion Notification Packets (CNPs) as congestion control messages.)
- **End-to-end congestion control leads to slow congestion response.** The ECN mechanism relies on congestion feedback received from the receiving end for congestion control. As can be seen from Fig. 1, when the transmission path of flow F1 gets congested, it takes at least one RTT time for the sender H1 to react to network congestion. During the congestion signal return path, the queue built on the bottleneck switch also significantly increases network latency, further exacerbating congestion.
- **Port-based ECN disregards fairness between queues.** The initial design of ECN only considered a single queue, but it is a general trend to have multiple queues per port of switches in modern data center networks. The variety of traffic types in data center networks has significantly increased. When the buffer of certain queues in a switch exceeds the configured threshold, all packets belonging to the same port may potentially receive ECN markings. In Fig. 1, at Time 0, H1 sends flow F1 of type A to H3. When congestion occurs at port P2 in node S2, packets at port P2 are marked with ECN. H3 responds with congestion feedback to reduce the sending rate of F1. At Time t, H0 sends

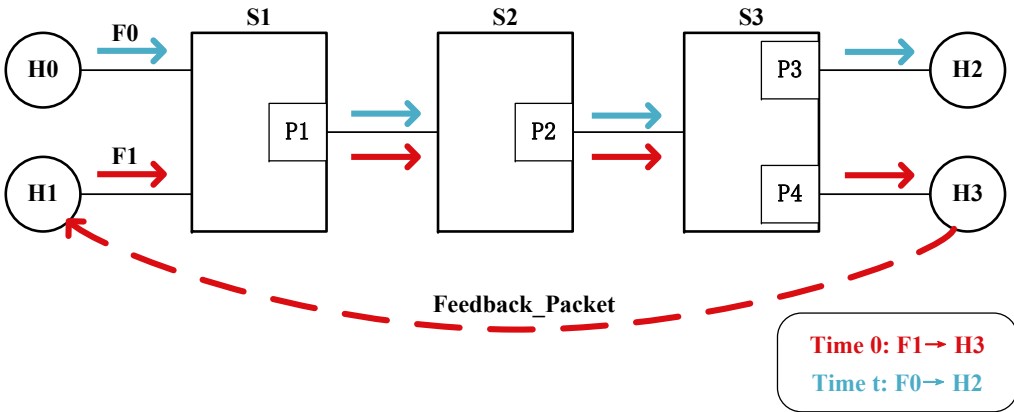

**Figure 1** **Diagram of end-to-end congestion control.** $H0$ and $H1$ are the sender, $H2$ and $H3$ are the receiver, $P1$, $P2$, $P3$, and $P4$ are the switch ports, and the dashed line indicates that the receiver returns the congestion signal to the sender.

flow F0 of type B to H2. If congestion at port P2 remains unresolved, flow F0 is also marked with ECN, thereby reducing the sending rate of H0. This implies that when the buffer of certain queues in a switch exceeds the considered threshold, the packets that are on the same port but going to a link that is not congested are marked, compromising the QoS for non-congested flows and disregarding fairness between queues.

To address the aforementioned issues, this project implements the ECN mechanism on a programmable data plane, leveraging the features of P4 to remove the constraints of protocols and devices on ECN markings. Simultaneously, a virtual queue mechanism is introduced, which divides the data streams into different queues within the switch. The corresponding queue ECN thresholds are set based on the QoS requirements of the traffic within each queue to ensure fairness of the traffic. Furthermore, a two-point architecture is employed to return congestion information at congested points, reducing the propagation delay of information. For the purpose of fulfilling the QoS requirements of the traffic within the queues, we found a multi-objective optimization model. Considering the model and solving requirements, a nonlinear programming problem-solving method is designed to solve the model. By employing this approach, we can guarantee the QoS requirements of the traffic while reducing network congestion issues.

## DESIGN

Compared to the traditional ECN mechanism that relies on the receiver informing the sender about congestion, this article aims to leverage the programmable data plane to directly return congestion information at congestion points (switches), thereby reducing message propagation time. In this article, the programmable capability of the data plane is utilized to generate feedback packets at switches. A data structure called "register" is employed to implement virtual queues within the switches. Additionally, data

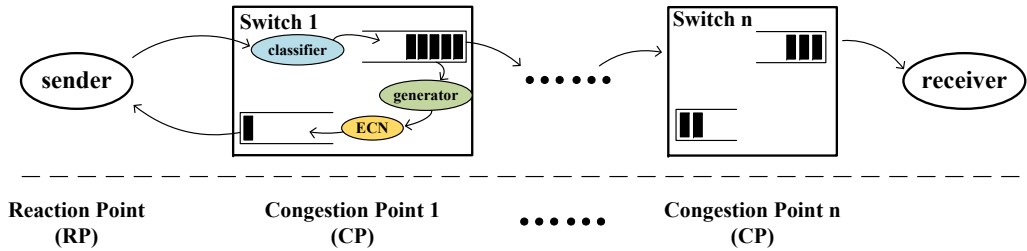

**Figure 2 Architectural model.** The sender is the reaction point (RP), and the intermediate switches are the congestion point (CP).

communication between the data plane and control plane is achieved through the Thrift interface of the programmable switch BMv2.

## ECN markup model—system architecture

The overall architecture of this mechanism in Fig. 2 is a dual-point congestion control architecture, consisting of the reaction point (RP) and the congestion point (CP). In contrast to the three-point architecture, the CP in the dual-point architecture is located on the switches and directly sends congestion feedback packets to the RP. Consequently, this architecture is more efficient than the three-point architecture as it reduces the round-trip delay of feedback packets.

In Fig. 2, the functions at the CP are divided into three parts: classifier, generator, and ECN marking. The classifier divides incoming packets into their respective virtual queues. When the length of a virtual queue exceeds a designated threshold, the generator creates corresponding feedback packets for that queue. The switch marks the ECN field in the feedback packets, writes congestion information into the feedback packets, and returns them to the source host to decrease the transmission rate of congested flows.

The RP point can adjust the packet sending rate of the source host. When congestion messages arrive, the source host adjusts the sending rate of the data flow based on the congestion information carried by the feedback packets. Additionally, when no congestion messages are received, the source host gradually increases the sending rate of the packets, allowing the rate to quickly recover to the pre-congestion level before the arrival of the congestion information, thereby increasing the link utilization.

## CP algorithm

As indicated by the CP in Fig. 3, this article deploys virtual queues in the data plane, enabling flexible traffic partitioning and queue scheduling. The control plane serves as the connection between the data plane and the multi-queue target optimization model, responsible for pushing the computed data down to the data plane. When packets arrive at the switch, they are partitioned into different virtual queues. The control plane calculates the optimal arrival rate and service rate for the current data flow based on the queuing theory model, taking into account the current queue conditions. It uses this information to compute the queue weights and the ECN marking thresholds, determining whether

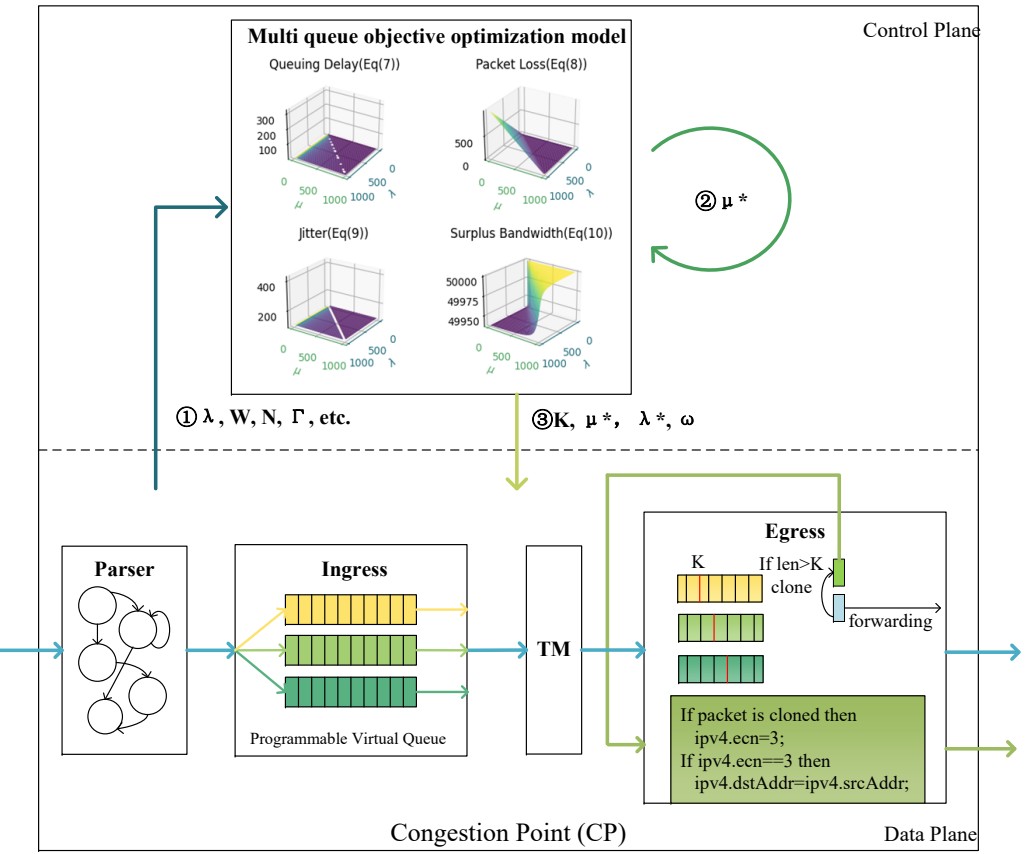

**Figure 3 Architecture diagram of CP points.** In control plane, the first step is to input parameters such as arrival rate $\lambda$, queueing delay $W$, total capacity of switches $N$, and total service rate $\Gamma$ of ports into the model. After obtaining the optimal service rate $\mu^*$ for the queue, the service rate $\mu^*$ is re-input into the model to derive the marking threshold $K$, optimal arrival rate $\lambda^*$, and queue weights $\omega$.

to generate ECN feedback packets. The marked feedback packets are returned to the source host, while the unmarked packets continue with subsequent packet scheduling and forwarding. This mechanism dynamically adjusts the arrival rate, service rate, queue weights, and ECN marking thresholds based on the current network traffic conditions, optimizing network performance and guaranteeing the QoS requirements of each traffic flow.

This marking strategy uses the last two bits of the IP header's ToS field to mark ECN as feedback packets, decoupling the ECN mechanism from the TCP protocol and switch configuration. When the queue length exceeds the ECN marking threshold $K$, the CP algorithm generates ECN feedback packets, the ECN bit of a feedback packet is marked as 11, indicating that the link where the packet resides is congested. Referring to the INT mechanism, packets contain INT headers to carry valid information. In the programmable data plane, the source IP is extracted from the packets arriving at the egress queue to serve as the destination IP for the feedback packets, and then the congestion packets are forwarded back to the source host.

Regarding the threshold, this article considers thresholds based on the QoS requirements of different flows, taking into account metrics such as throughput, latency, and packet loss, and treats them as optimization objectives. The research simultaneously considers the relationship between marking thresholds, queue weights, and QoS objectives, describing the optimization objectives as a non-linear programming problem and establishing a multi-QoS optimization model. The model is solved using optimization theory to obtain optimal marking thresholds and queue weights, thus achieving the QoS requirements of different flows. Specifically, Algorithm 1 describes the entire process of packets within the switch.

---

**Algorithm 1** The CP algorithm

---

**Input:** packet;
**Output:** packet, feedback packet;
  **while** Packet_in **do**
    Partition the packets into virtual queue $i$ while collecting packet information such as arrival rate $\lambda_i$, queuing delay $W_i$, etc.
    Pass the collected information to the control plane to obtain the queue ECN marking threshold $K_i$, optimal arrival rate $\lambda_i^*$, and queue weight $\omega_i$.
    **while** the number of packets with priority $P > \omega_i$ **do**
      $P--$;
    **end while**
    packet.priority = $P$;
    **if** $Q\_length > K_i$ **then**
      clone the packet;
    **end if**
    **if** the packet is a clone packet **then**
      packet.IP.tos=3;
      packet.INT.rate1=$\lambda_i$;
      packet.INT.rate2=$\lambda_i^*$;
    **end if**
    Forward.
  **end while**

---

### Virtual queue and queue scheduling

This article utilizes BMv2 switches for programmable data plane simulation. By default, BMv2 switches employ Strict Priority (SP) queues, where packets are assigned to different queues based on their priority, enabling multi-queue emulation. However, SP queues neglect fairness among queues. In the event of congestion, if there are continuously packets in the high-priority queue, packets in the low-priority queue will not receive service. Additionally, due to the developmental stage of BMv2 switches, some functionalities are incomplete, and obtaining queue parameters for individual subqueues is not possible

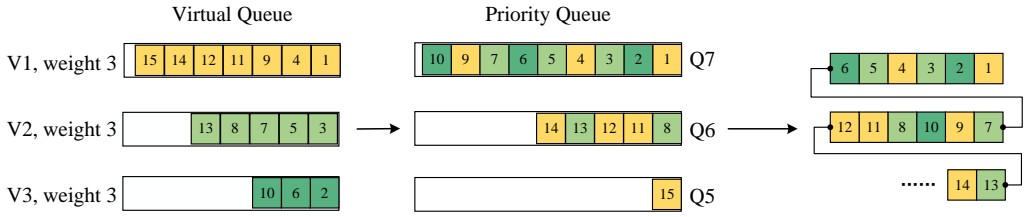

**Figure 4** **Mapping from virtual queue to priority queue.** The last part is the order of the packets leaving the switch.

(*Harkous et al., 2021*). To address these limitations, this article proposes a virtual queue-based Weighted Round-Robin (WRR) scheduling mechanism, which is based on priority queues, to ensure fairness among queues. This approach aims to alleviate the fairness issues associated with SP queues, while also overcoming the incomplete functionality of BMv2 switches in obtaining subqueue parameters.

In our P4 pipeline, we utilize virtual queues, which are mechanisms used to model the length of queues. Unlike regular queues, virtual queues do not contain any actual packet data. Instead, they are numeric values that increment as packets arrive and decrement based on a predefined model. In our implementation, we utilize the data structure register available in the P4 language to implement virtual queues and monitor their states. The register allows us to store and manipulate the values associated with each virtual queue, providing a way to track and manage their state within the P4 pipeline.

We employ a matching table to divide each incoming packet into different virtual queues based on their source and destination addresses. Furthermore, we assign weights to these virtual queues and utilize the register to track the utilization of the priority queues within each virtual queue. As depicted in Fig. 4, if the utilization of a priority queue within a specific virtual queue exceeds the weight associated with that virtual queue, we enforce the usage of a lower priority queue. In the algorithm designed in this article, the queue weights are controlled by the control plane.

In Fig. 4, the sequence number on the packets represents the order in which they arrive at the switch. With a queue weight set to 3, when a packet arrives at a virtual queue, the first-arrived packet in that virtual queue is assigned the highest priority. If, for a particular virtual queue, the length of its packets in a priority queue reaches 3, the priority of that packet is downgraded, and it enters a lower-priority queue. This process continues, enabling packet forwarding according to the WRR scheduling method.

To validate that simulating WRR with virtual queues can mitigate the impact of SP queues on traffic in BMv2 switches, this experiment ran both the default SP scheduler and a WRR scheduler based on virtual queues in the switch. High, medium, and low priority traffic were simultaneously sent to the switch with equal weights for the three types of traffic. The bandwidth performance of each traffic type under the two scheduling schemes is shown in Table 1. It can be observed that in SP scheduling, low-priority packets have fewer scheduling opportunities, resulting in lower bandwidth. In contrast, with WRR scheduling, this issue is addressed, and all three types of traffic have opportunities to transmit data.

**Table 1  Bandwidth of three types of traffic (Mbps).**

| Scheduling | High | Medium | Low |
|---|---|---|---|
| SP | 6.222333333 | 3.565245902 | 0.1583333 |
| WRR | 3.090322581 | 3.017741935 | 2.992786885 |

Therefore, in the programmable data plane, WRR scheduling based on virtual queues can effectively mitigate the problem of flow starvation compared to the default SP scheduling, ensuring fairness among traffic flows.

### Model establishment

In this chapter, according to reference (*Wu, Qiao & Chen, 2018*), the data flows in a network can be broadly classified into the following three categories to meet the QoS requirements of different types of data streams:

**Category A data flows:** These are bandwidth-sensitive and delay-sensitive data streams. This category of data flows has higher requirements for network performance, necessitating sufficient bandwidth and low transmission latency. Typically, these data flows include applications such as audio, video, and real-time gaming that have high demands for real-time transmission. Ensuring priority transmission for these data flows ensures their immediate and seamless delivery in the network, providing a good user experience and service quality.

**Category B data flows:** These are delay-sensitive data streams. This category of data flows is sensitive to transmission latency, and the network should provide low-latency transmission for them.

**Category C data flows:** These are best-effort data flows, which do not have specific requirements for latency and bandwidth. A common example is file transfer protocol (FTP) traffic. For these data flows, the main objective of the network is to deliver the data to the best of its ability.

By classifying data flows into different categories, networks can employ appropriate strategies and mechanisms to provide corresponding quality of service guarantees for each category of data flows, meeting the requirements and performance needs of diverse applications. The symbols and meanings used in this article are shown in Table 2.

We define $I = \{1, 2, \ldots, k\}$ to represent the set of traffic types present in the network, where traffic $i, i \in I$ represents a specific type of flow. When traffic arrives at the switch, the classification module will allocate traffic $i$ to the corresponding queue, and the ECN marking module will set the appropriate ECN marking threshold in accordance with the business requirements of traffic $i$. Let the optimized service rate be used as the threshold $K_i$. Assuming that each switch has the same capacity limit and can accommodate a queue length of $N$, the arrival rate of traffic $i$ is $\lambda_i$, and the service rate is $\mu_i$. Let $\Lambda$ represent the total arrival rate of packets at the switch, and $\Gamma$ represent the total service rate of packets leaving the switch through the same port. All packets, after passing through the classifier, are stored in different queues, and only one scheduler serves multiple queues. Let $\omega_i$ represent the weight of traffic $i$'s queue in the virtual queue, and $\sum_{i=1}^{k} \omega_i = 1$. The weight determines how many packets from that queue are served by the scheduler in one unit of

**Table 2  Model symbols and parameters.**

| Notation | Description |
|---|---|
| $\lambda_i$ | The arrival rate of traffic type $i$. |
| $\mu_i$ | The service rate of traffic type $i$. |
| $K_i$ | The ECN marking threshold for the queue containing traffic $i$. |
| $N$ | The number of packets that can be accommodated in each switch. |
| $\omega_i$ | The proportion of the queue in which traffic type $i$ resides in relation to all queues in the virtual queue. |
| $\Lambda$ | The total arrival rate of packets arriving at the switch. |
| $\Gamma$ | The aggregate service rate of packets departing from the switch on the same port. |
| $P_m$ | The probability that there are $m$ packets in queue $i$ at time $t$. |
| $L_q^i$ | Expected value of the number of packets waiting in the queue containing traffic $i$. |
| $L_c^i$ | Expected value of the number of packets receiving services in the queue containing traffic $i$. |
| $W_q^i$ | Queueing time of packets in the queue containing traffic $i$. |
| $W_c^i$ | Packet service time of the queue containing traffic $i$. |
| $C$ | The bandwidth of each individual link. |
| $n$ | The number of links connected to the switch. |
| $W_i$ | The actual queuing time of traffic type $i$ within the switch. |
| $\lambda_i^*$ | The optimal arrival rate of the queue containing traffic $i$. |
| $\mu_i^*$ | The optimal service rate of the queue containing traffic $i$. |

time. If the scheduler serves a total of $\Gamma$ packets in one unit of time, then $\omega_i \Gamma$ packets are served in queue $i$, *i.e.*, $\mu_i = \omega_i \Gamma$. Therefore, the weighted round-robin scheduling queue model can be viewed as k parallel M/M/1/N queuing models.

In the multiple parallel queues of M/M/1/N, it is sufficient to consider only one of them because their only difference lies in the input rate and service rate. In the following discussion, we will focus on the queue associated with traffic $i$.

$P_m$ represents the probability that there are $m$ packets in the queue where traffic $i$ resides at time $t$. We can use the method of discrete-time Markov chains to establish the probability distribution of queue length. For the M/M/1/N queuing model, assuming the steady-state distribution of queue length is $P_m$, based on the principles of queuing theory, we can derive the following equations:

$$P_0 = \frac{1-\rho_i}{1-\rho_i^{N+1}}, \rho \neq 1 \tag{1}$$

$$P_m = \frac{1-\rho^i}{1-\rho_i^{N+1}} \rho_i^m, 1 \leq m \leq N \tag{2}$$

where $\rho_i = \lambda_i / \mu_i$. In the system with a capacity of $N$, when the queue length reaches $N$, any additional arriving packets beyond the capacity will be discarded. Consequently, for queue $i$, its effective arrival rate can be determined using the following equation, as indicated in Eq Eq. (3), where $P_N$ represents the probability of the queue length reaching the upper limit $N$.

$$\lambda_e = \lambda_i(1 - P_N). \tag{3}$$

Through the decomposition of parallel queues, we can determine the relationship between the arrival rate $\lambda_i$ and service rate $\mu_i$ of flow $i$ with the parameters $\Lambda$ and $\Gamma$ of the overall parallel queue system.

$$\sum_{i=1}^{k} \lambda_i \leq \Lambda, \sum_{i=1}^{k} \mu_i = \Gamma. \tag{4}$$

So, as to the stability of the network, there should be:

$$\lambda_i \leq \mu_i, \sum_{i=1}^{k} \lambda_i \leq \Gamma. \tag{5}$$

When $\rho_i = \lambda_i / \mu_i \neq 1$, the calculated metric results are as follows:

**(1) Queue length:** The average length $L_s^i$ of queue $i$ is equal to the sum of the expected value $L_q^i$ of the number of packets waiting to queue in the system and the expected value $L_c^i$ of the number of packets being served.

$$L_q^i = \sum_{k=0}^{N-1} k P_{k+1} = \frac{\rho_i}{1 - \rho_i} - \frac{N\rho_i^{N+1} + \rho_i}{1 - \rho_i^{N+1}}$$

$$L_c^i = 1 - P_0 = \frac{\rho_i - \rho_i^{N+1}}{1 - \rho_i^{N+1}} \tag{6}$$

$$L_s^i = L_q^i + L_c^i = \frac{\lambda_i}{\mu_i - \lambda_i} - \frac{(N+1)\lambda_i^{N+1}}{\mu_i^{N+1} - \lambda_i^{N+1}}.$$

**(2) Queueing delay:** The waiting time $W_s^i$ of packets in queue $i$ is equal to the sum of the queuing time $W_q^i$ and the service time $W_c^i$.

$$W_s^i = W_q^i + W_c^i = \frac{L_q^i}{\lambda_e} + \frac{L_c^i}{\lambda_e}$$

$$W_s^i = \frac{1}{\mu_i - \lambda_i} - \frac{N\lambda_i^N}{\mu_i(\mu_i^N - \lambda_i^N)}. \tag{7}$$

**(3) Packet loss count:** When the queue capacity of a switch is reached ($N$ packets), any new incoming packets will be dropped. Therefore, the packet loss count in the system per unit of time is determined.

$$N_{loss} = \lambda_i P_N = \frac{\lambda_i^{N+1}(\mu_i - \lambda_i)}{\mu_i^{N+1} - \lambda_i^{N+1}}. \tag{8}$$

**(4) Queueing delay jitter:** We define jitter as the difference between the actual delay experienced by a packet in the switch and the expected delay. Let $W_i$ denote the actual queueing time for traffic flow $i$ in the switch. The jitter of the queueing delay, compared to the expected queueing delay, can be represented by the following equation:

$$\triangle T = |W_i - W_s^i|. \tag{9}$$

**(5) Link remaining bandwidth:** Assuming $C$ is the bandwidth of every link and $n$ represents the number of links connected to the switch, the available remaining bandwidth of the link per unit of time can be expressed as the difference between the total bandwidth and the amount of data transmitted to the switch per unit of time (bandwidth occupied by existing traffic).

$$C_{rest} = nC - \sum_{i=1}^{k} L_s^i = nC - \sum_{i=1}^{k} \left[ \frac{\lambda_i}{\mu_i - \lambda_i} - \frac{(N+1)\lambda_i^{N+1}}{\mu_i^{N+1} - \lambda_i^{N+1}} \right]. \tag{10}$$

We assume the presence of three types of traffic services in the network: voice, video, and email. Real-time data, such as voice, imposes high demands on both delay and bandwidth. Streaming applications also have certain requirements on latency as they employ buffering mechanisms and can tolerate reduced bandwidth to some extent compared to real-time applications. File transfer services, such as email, generally have lower bandwidth and latency requirements.

Taking voice traffic as an example, based on the QoS requirements for voice flows, we develop an optimization model that jointly adjusts the arrival rate and service rate of flows. The objective is to minimize delay and residual bandwidth by optimizing the service rate and target sending rate.

$$min(C_{rest}) = min(nC) - \sum_{i=1}^{k} \left[ \frac{\lambda_i}{\mu_i - \lambda_i} - \frac{(N+1)\lambda_i^{N+1}}{\mu_i^{N+1} - \lambda_i^{N+1}} \right]$$

$$min(W_s^i) = min(\frac{1}{\mu_i - \lambda_i} - \frac{N\lambda_i^N}{\mu_i(\mu_i^N - \lambda_i^N)})$$

$$s.t.$$

$$C1 : \sum_{i=1}^{k} \omega_i = 1, \mu_i = \omega_i \Gamma, \omega_i > 0$$

$$C2 : \sum_{i=1}^{k} L_s^i = \sum_{i=1}^{k} (\frac{\lambda_i}{\mu_i - \lambda_i} - \frac{(N+1)\lambda_i^{N+1}}{\mu_i^{N+1} - \lambda_i^{N+1}}) \leq N \tag{11}$$

$$C3 : \sum_{i=1}^{k} \lambda_i \leq \Lambda$$

$$C4 : 0 < \lambda_i < \mu_i$$

$$C5 : \sum_{i=1}^{k} \mu_i = \Gamma$$

$$C6 : C_{rest} = nC - \sum_{i=1}^{k} \left[ \frac{\lambda_i}{\mu_i - \lambda_i} - \frac{(N+1)\lambda_i^{N+1}}{\mu_i^{N+1} - \lambda_i^{N+1}} \right] \geq 0.$$

$C1$ represents the relationship between $\mu_i$, $\omega_i$, and $\Gamma$. $C2$ states that the sum of average queue lengths in all queues should not exceed the storage capacity of the switch. $C3$ states that the number of packet arrivals for all traffic flows within a unit of time should not exceed the number of packet arrivals at the port within the same unit of time. $C4$ states that in order to maintain network stability, the arrival rate of traffic should not exceed the service rate. $C5$ states that the number of packets processed for all traffic flows within a unit of time should be equal to the number of packets processed at the port within the same unit of time. C6 states that the available remaining bandwidth of a link should be greater than zero.

Video streams are sensitive to both delay and jitter, so their optimization objective is to minimize latency and jitter.

$$min(W_s^i) = min(\frac{1}{\mu_i - \lambda_i} - \frac{N\lambda_i^N}{\mu_i(\mu_i^N - \lambda_i^N)})$$
$$min(\triangle T) = min(|W_i - W_s^i|). \tag{12}$$

File transfers are sensitive to both delay and packet loss, so their optimization objective is to minimize latency and packet loss rate.

$$min(W_s^i) = min(\frac{1}{\mu_i - \lambda_i} - \frac{N\lambda_i^N}{\mu_i(\mu_i^N - \lambda_i^N)})$$
$$min(N_{loss}) = min(\frac{\lambda_i^{N+1}(\mu_i - \lambda_i)}{\mu_i^{N+1} - \lambda_i^{N+1}}). \tag{13}$$

The constraints of the above optimization target models are the same.

### The optimal arrival rate, service rate, and virtual queue weights

In the aforementioned mathematical model, this study aims to determine the optimal arrival rate, $\lambda_i^*$, and optimal service rate, $\mu_i^*$, given the optimization objective. To achieve this, a two-layered solving approach is devised. Firstly, the actual arrival rate, $\lambda_i$, is inputted into the model to solve for the optimal service rate, $\mu_i^*$, thereby completing the first optimization layer. Subsequently, the optimal service rate, $\mu_i^*$, is fed back into the model to solve for the optimal arrival rate, $\lambda_i^*$, thereby completing the second layer of optimization. It is worth noting that both layers of the solving model are multi-objective non-linear models, necessitating consistency in the solving method for each layer.

First, we can apply the method of linear weighted sum in multi-objective optimization. Assuming the weight coefficients for the three traffic flows are $\omega_1$, $\omega_2$, and $\omega_3$, respectively, with $\omega_i = \mu_i / \sum \mu_i$, the above multi-objective optimization problem can be transformed into a single-objective optimization problem by linearly weighting the objectives. Let $f(x)$ represent the overall optimization objective for traffic:

$$f(x) = min(\omega_1 Eq(11) + \omega_2 Eq(12) + \omega_3 Eq(13)). \tag{14}$$

Therefore, the current focus is to solve a single-objective multi-variable nonlinear programming problem.

Differential evolution (DE) is a stochastic search algorithm designed specifically for global optimization problems. It originated from the early development of genetic

algorithms (GA). Differential evolution employs a unique evolutionary mechanism that utilizes the individual differences within the existing population to construct mutant individuals. Compared to other intelligent algorithms such as genetic algorithms, particle swarm optimization, ant colony optimization, and artificial bee colony algorithms, DE requires fewer parameters and exhibits strong optimization capability. DE belongs to the category of evolutionary algorithms and inherits all the advantages of evolutionary algorithms. The DE algorithm selects a more optimal next generation through processes such as population initialization, individual fitness evaluation, differential mutation operation, crossover operation, and selection operation. In summary, the DE algorithm is suitable for solving large-scale, nonlinear, and combinatorial optimization problems that are challenging for traditional search methods. It does not rely on gradient information, nor does it require the objective function to be continuous or differentiable.

In this study, the DE/best/1/L (*Opara & Arabas, 2019*) algorithm is adopted. The population is classified and described, and the target-to-best strategy is used to select the base individuals for differential mutation from the current population. Differential mutation is applied to the current population to generate mutant individuals. Then, the current population is combined with the mutant individuals to obtain an experimental population using the exponential exchange method. A one-to-one survivor selection is performed between the current population and the experimental population to generate a new population. The algorithm for determining the optimal packet arrival rate $\lambda_i^*$, service rate $\mu_i^*$, and virtual queue weight $\omega_i$ is as follows (Algorithm 2):

## RP algorithm

The RP algorithm is deployed on the end hosts in the network, and the rate variation pattern is shown in Fig. 5. Feedback packets are sent by the CP and carry ECN information and switch information. Upon receiving the ECN feedback packet, RP reduces the transmission rate of traffic sent to the network and records this rate as the target rate for fast recovery before receiving the feedback information. The feedback packet information includes the optimized destination switch's optimal arrival rate and the original arrival rate. Since the sending rate is directly proportional to the arrival rate, we can establish the following relationship, where $\lambda^*$ represents the optimal arrival rate, $\lambda$ represents the original arrival rate, $v^*$ represents the adjusted sending rate, and $v$ represents the original sending rate.

$$\frac{\lambda^*}{\lambda} = \frac{v^*}{v}. \tag{15}$$

Therefore, the formula for reducing the rate is as follows: $R_t$ is the target rate of fast recovery stage, $R_{old}$ is the rate before receiving the feedback packet, and $R_c$ is the current rate.

$$
\begin{aligned}
R_t &= R_{old} \\
R_c &= \frac{\lambda^*}{\lambda} R_c \\
R_d &= R_t - R_c.
\end{aligned}
\tag{16}
$$

---

**Algorithm 2** The optimal arrival rate, service rate, and virtual queue weight algorithm

---

**Input:** $N, \Lambda, \Gamma, C, n, \lambda_i, W_i$ and other model parameters;

**Output:** $\lambda_i^*, \mu_i^*, \omega_i$;

**Step 1:** Initialize;

**Step 2:** The multi-objective optimization problem is transformed into a single-objective optimization problem by linear weighted sum method, Let $f_i(x)$ represent the optimization objective for traffic flow $i$:

$min(\frac{\mu_1}{\Gamma}f_1(\mu_1) + \frac{\mu_2}{\Gamma}f_2(\mu_2) + \frac{\mu_3}{\Gamma}f_3(\mu_3))$

$s.t.$

$C1 : \sum_{i=1}^{k} L_s^i \leq N$

$C2 : 0 < \lambda_i < \mu_i$

$C3 : \sum_{i=1}^{k} \mu_i = \Gamma$

$C4 : C_{rest} = nC - \sum_{i=1}^{k}\left[\frac{\lambda_i}{\mu_i - \lambda_i} - \frac{(N+1)\lambda_i^{N+1}}{\mu_i^{N+1} - \lambda_i^{N+1}}\right] \geq 0$

**Step 3:** Find the optimal solution $\mu_i^*$ by differential evolution algorithm, the virtual queue weight is $\frac{\mu_i^*}{\mu_1 + \mu_2 + \mu_3}\Gamma$, and the queue marking threshold $K_i = \frac{\mu_i^*}{\mu_1 + \mu_2 + \mu_3}N$;

**Step 4:** Construct the second layer single objective optimization problem:

$min(\frac{\mu_1^*}{\Gamma}f_1(\lambda_1) + \frac{\mu_2^*}{\Gamma}f_2(\lambda_2) + \frac{\mu_3^*}{\Gamma}f_3(\lambda_3))$

$s.t.$

$C1 : \sum_{i=1}^{k} L_s^i \leq N$

$C2 : \sum_{i=1}^{k} \lambda_i \leq \Lambda$

$C3 : 0 < \lambda_i < \mu_i^*$

$C4 : C_{rest} = nC - \sum_{i=1}^{k}\left[\frac{\lambda_i}{\mu_i - \lambda_i} - \frac{(N+1)\lambda_i^{N+1}}{\mu_i^{N+1} - \lambda_i^{N+1}}\right] \geq 0$

**Step 5:** Input model parameters such as $N, \Lambda, \Gamma, C, n, \lambda_i, W_i$ and the optimal solution $\mu_i^*$

**Step 6:** Find the optimal solution $\lambda_i^*$ by differential evolution algorithm.

---

The source host restricts its transmission rate according to the feedback packets received from the CP. However, when no feedback packet is received, the host enters the fast recovery phase and gradually increases the transmission rate to recover the previously reduced rate. The rate increase follows Eq. (17), where $R_t$ remains constant and $R_c$ is updated according to the equation.

$$R_c = R_c + \frac{R_d}{2} = \frac{R_c + R_t}{2}. \tag{17}$$

The fast recovery phase lasts for $N_C$ cycles, and in P4QCN, it is recommended to set the value of $N_C$ as 5 (*Geng, Yan & Zhang, 2019*). Upon completion of these $N_C$ cycles, the system transitions to the active increase phase to discover any available excess bandwidth. The sending rate variation for the RP is as follows, where $R_A$ represents a constant, which is set to 5 Mbps in the case of P4QCN.

$$\begin{aligned} R_t &= R_t + R_A \\ R_c = \frac{R_c + R_t}{2} &= \frac{R_c + R_t + R_A}{2}. \end{aligned} \tag{18}$$

The overall process of the RP algorithm is illustrated by Algorithm 3.

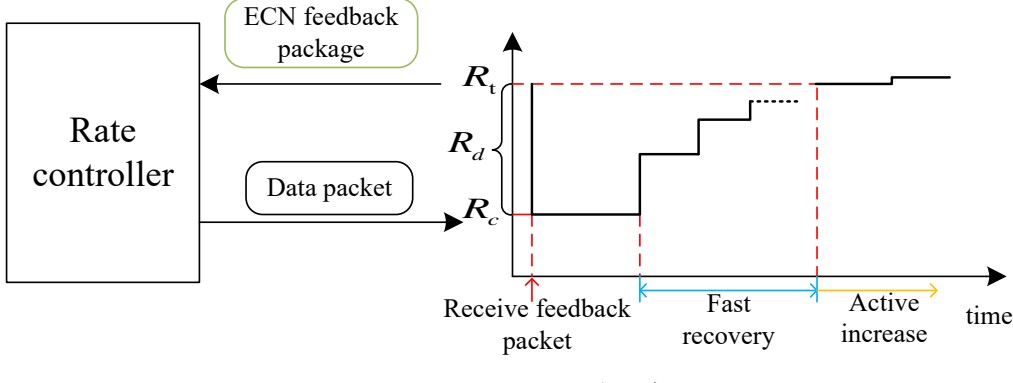

Reaction Point(RP)

**Figure 5** **Plot of the rate change at the RP point.** On the left is the rate controller and on the right is the rate change diagram. Rate changes can be divided into three stages: speed reduction (when feedback packets are received), fast recovery, and active increase. $R_t$ represents the target rate for fast recovery, $R_c$ is the current rate, and $R_d$ is the difference between the two.

# EXPERIMENT

The development and testing of experiments in this section were conducted on a server running Ubuntu 18.04 64-bit. Network simulation utilized a P4 programmable simulation environment constructed jointly by Mininet and BMv2 software switches. The experiments in this article utilized the P4 version P6_16. We implemented ECN marking, feedback packet generation, and weighted round-robin packet scheduling mechanism using the P4 language in the BMv2 switch. We conducted comparative experiments with DCQCN (*Zhu et al., 2015*), P4QCN (*Geng, Yan & Zhang, 2019*), and TCN (*Bai et al., 2016a*) to evaluate the performance of this strategy. We chose DCQCN, P4QCN, and TCN as the comparative experiments because they are the primary congestion control mechanisms currently used in data center networks. Among them, DCQCN is a static ECN marking strategy, P4QCN is a two-point architecture, and TCN is a dynamic ECN marking mechanism based on multiple queues and delay. The configuration of relevant parameters of the experiment is shown in Table 3:

Due to the simulation environment performance limitation, we set each link bandwidth to 10Mbps in the testbed. We implemented the DCQCN, P4QCN and TCN mechanisms based on the algorithms on the testbed, with the ECN labeling threshold $K_{DCQCN} = K_{P4QCN} = 11$, and for TCN, we used the parameter $\lambda = 0.17$ suggested in the literature (*Kim & Lee, 2021*; *Zhang, Bai & Chen, 2019*). We constructed the topology structure shown in Fig. 6 using Mininet, hosts $H1, H2, H3$ are the sender, $H4, H5$ are the receiver, we use python's Scapy library at the sender to implement the RP algorithm, and obtain network performance parameters including bandwidth, RTT and total queue length within the switch through INT mechanism. We design two packet delivery scenarios and contrast the performance of each strategy in these two scenarios. In the following

---

**Algorithm 3** The RP algorithm

---

**Input:** State, $R_A = 5Mbps$, $send\_rate = 1Mbps$, $R_t = 0$, $N_C$=5, $\lambda_i, \lambda_i^*$;

**Output:** $send\_rate$;

  State = active increase;

  **while** not receive feedback packet **do**

    **if** State == active increase **then**

      $send\_rate = \frac{send\_rate + R_t + R_A}{2}$;

    **end if**

    **if** State == fast recovery **then**

      $send\_rate = \frac{send\_rate + R_t}{2}$;

      $N_C = N_C - 1$;

      **if** $N_C == 0$ **then**

        State = active increase;

      **end if**

    **end if**

  **end while**

  **while** receive feedback packet **do**

    State = rate reduction;

    $R_t = send\_rate$;

    $send\_rate = \frac{\lambda_i^*}{\lambda_i} send\_rate$;

    State=fast recovery;

  **end while**

---

**Table 3  Configure settings and related parameters.**

| Operating system | Ubuntu 18.04 64-bit |
|---|---|
| Language | Python 3.8, P4_16 |
| Threshold value | $K_{DCQCN} = K_{P4QCN} = 11$ |
| λ of TCN | 0.17 |
| Link bandwidth | 10Mbps |

tests, the strategy proposed by us is named PMQECN (Programmable Multi-QoS Explicit Congestion Notification) for the convenience of image reading.

## Scenario 1: Performance test

**Scenario 1**: In this scenario, $H1, H2$, and $H3$ simultaneously send Class A, Class B, and Class C packets to $H5$. The initial sending rate at the RP point is set to 1Mbps, and the initial phase follows an actively increasing pattern. As the sending rate increases, congestion occurs in the link. The receiving end, $H5$, captures INT packets to obtain network status. We evaluate the performance from three aspects: queue length inside the switches, round-trip time (RTT), and bandwidth.

    Based on INT packets, we gather information about the queue length inside the switches, as shown in Fig. 7. In the absence of congestion, the packet count gradually increases over

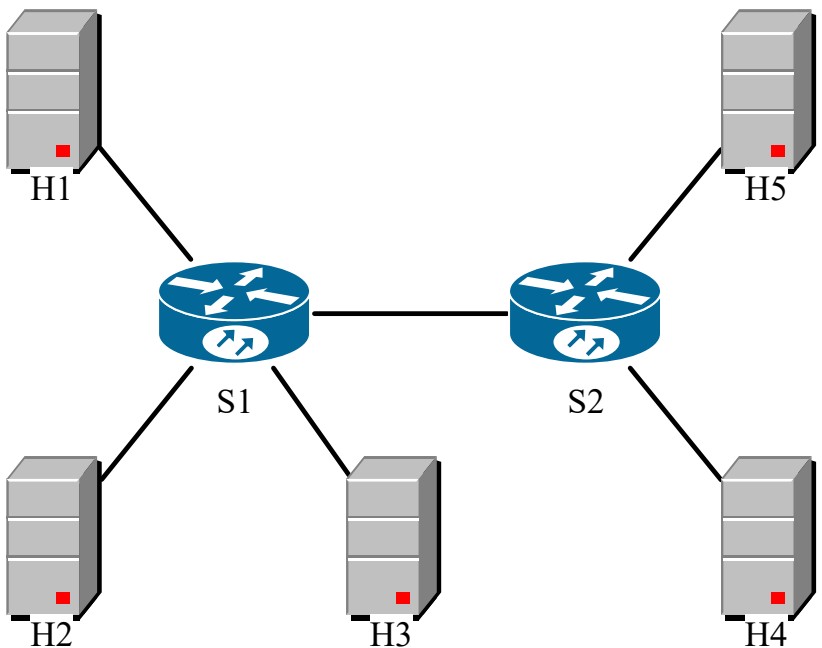

**Figure 6** **Topological structure.**

time. Once the congestion threshold is reached, our algorithm adjusts the threshold in the real time on the basis of the internal switch conditions and promptly returns congestion feedback packets to reduce the queue length within the switch. In Fig. 7, it can be seen that our algorithm makes the internal queue length of the switch significantly lower than DCQCN, P4QCN and TCN, and the maximum queue length is reduced by 56%, 31% and 17%, respectively. DCQCN and P4QCN, which adopt fixed thresholds, experience significant fluctuations in queue length. DCQCN, operating under the RP-CP-NP architecture, has a higher delay for the feedback information to reach the source host compared to P4QCN. Hence, the reaction time for speed reduction is slightly delayed in DCQCN. TCN dynamically adjusts the threshold based on packet delay, resulting in relatively stable overall variations. However, similar to DCQCN, the reaction time at the sender is slower in TCN.

We obtain the packet's arrival delay through the timestamp information carried by the packet and estimate the RTT by doubling this arrival delay. As shown in Fig. 8, DCQCN exhibits significantly higher delays compared to the other three methods. TCN, which utilizes packet delay as the ECN marking threshold, maintains relatively stable performance in terms of packet RTT. Overall, both P4QCN and our proposed algorithm demonstrate good performance in terms of RTT. The average delay information obtained through computation indicates that the strategy proposed in this article reduces the delay by 89.4%, 5.3%, and 47% compared to DCQCN, P4QCN, and TCN, respectively. Notably, both

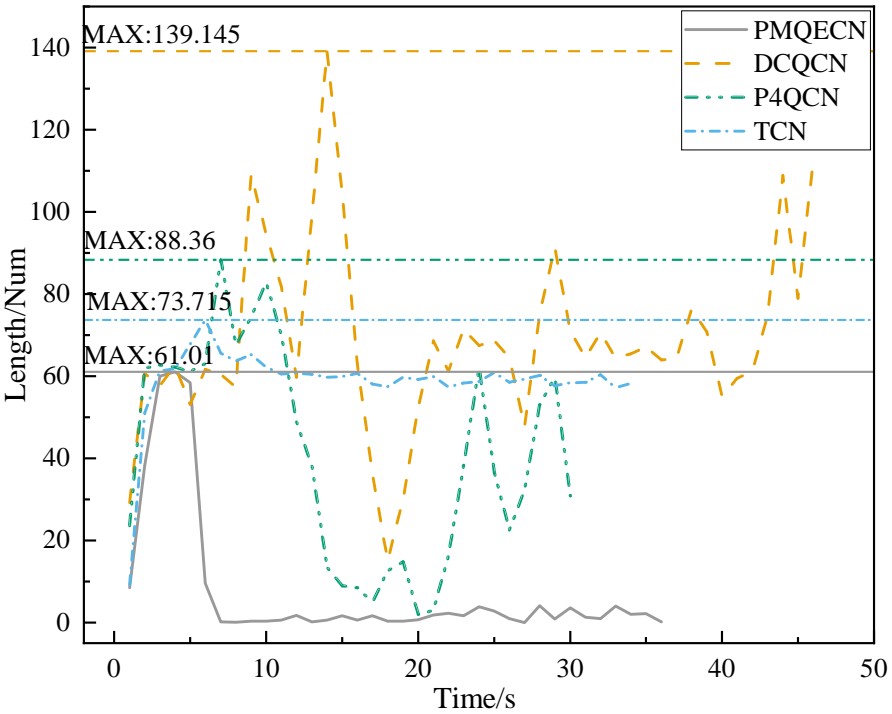

**Figure 7** Scenario 1: Comparison of the queue length of PMQECN, P4QCN, DCQCN, and TCN on the switch.

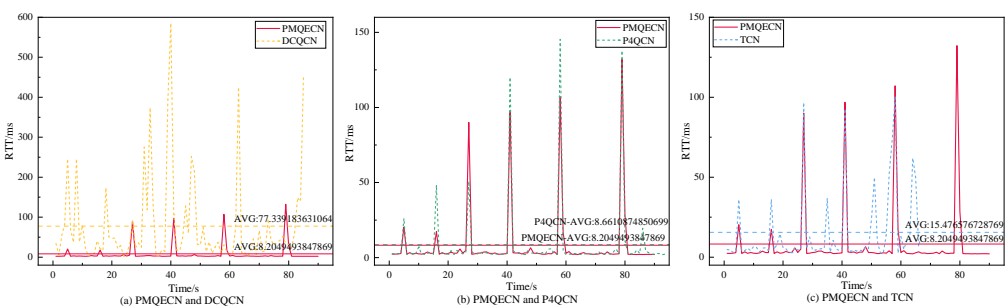

**Figure 8** Scenario 1: Comparison of round trip latency of PMQECN, DCQCN, P4QCN, and TCN in links.

this strategy and P4QCN employ a two-point architecture. It is evident that a two-point architecture, compared to a three-point architecture, effectively reduces network latency.

We measured the throughput received by the $H5$ port as shown in Fig. 9. The link information is recorded for every packet received by the $H5$ port. After a period of reception, we calculated the average throughput per second over 60 s to depict the variation in port throughput within that timeframe. Excluding unstable data at the beginning and end, we selected the middle portion as our sample data (this is because the header and tail data correspond to the initial and final packets sent. We consider this portion of the data to be

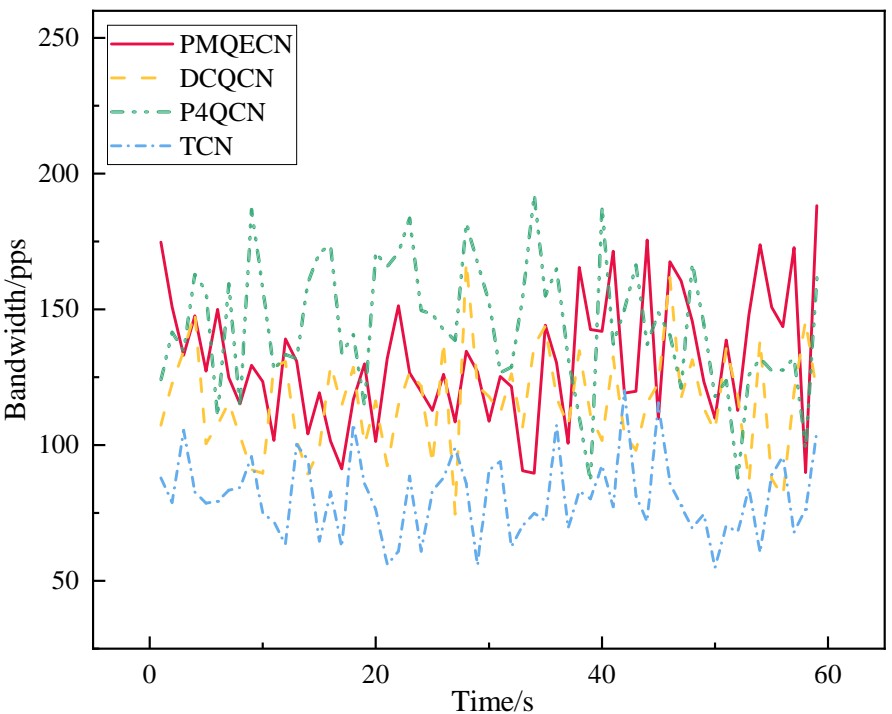

**Figure 9** Scenario 1: Comparison of port throughput of PMQECN, P4QCN, DCQCN, and TCN on $H$5.

more random and significantly different from the data obtained during the steady operation of the transmission system. Therefore, we selected the middle portion of the data for analysis.). We computed the standard deviations and coefficients of variation (CV) under steady-state conditions for several algorithms, as depicted in Fig. 10. Relative to DCQCN, P4QCN, and TCN, our algorithm exhibits a smaller CV, indicating less relative variability in its numerical values. P4QCN, due to its two-point architecture, achieves faster feedback, but its fixed threshold approach results in relatively large fluctuations in throughput. During testing, we observed instances where throughput sharply dropped to minimum levels before gradually recovering over time. While P4QCN's CV is smaller compared to DCQCN and TCN, its large standard deviation suggests a higher mean throughput for P4QCN. TCN shows a high CV but a small standard deviation, typically indicating a lower mean throughput with data points more tightly clustered around the mean but with larger differences between individual data points. Our algorithm incorporates dynamic threshold feedback atop rapid feedback mechanisms to reduce throughput fluctuations while ensuring throughput levels. To ensure data representativeness, statistical parameters such as sample means, confidence intervals, and overall population means are presented in Table 4. Each algorithm's overall mean falls within the 95% confidence interval, indicating high accuracy in our estimation of the population mean based on sample data.

We utilized Wireshark to gather statistics on the overall packet loss rate and the packet loss rates for individual traffic classes, as depicted in Fig. 11. Class A, Class B, and Class C represent different types of packets transmitted by H1, H2, and H3, respectively. From
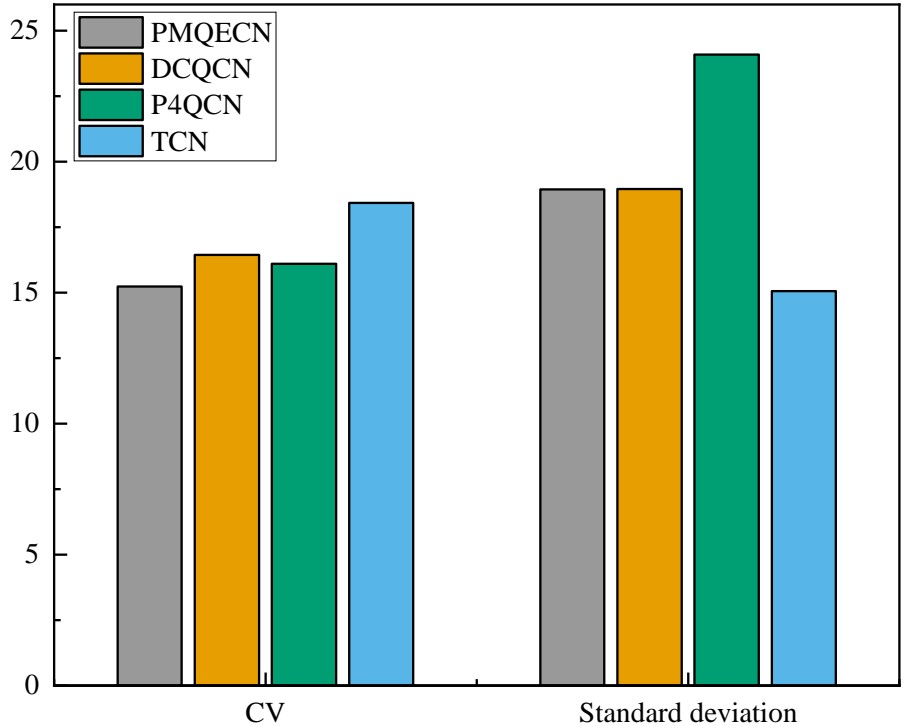

**Figure 10** Scenario 1: Standard deviation and CV of throughput for PMQECN, DCQCN, and P4QCN.

**Table 4** Table of statistical parameters.

| Algorithm | Mean of the sample | Standard deviation of the sample | CV | 95% confidence interval | Mean of the population |
|---|---|---|---|---|---|
| PMQECN | 124.314 | 18.94446 | 0.15239 | (118.33442, 130.29363) | 130.0919 |
| DCQCN | 115.3662 | 18.96285 | 0.16437 | (109.38075, 121.35158) | 116.2216 |
| P4QCN | 149.6058 | 24.08689 | 0.161 | (142.00299, 157.20852) | 143.6532 |
| TCN | 81.66243 | 15.05175 | 0.18432 | (76.91151, 86.41335) | 80.63305 |

Fig. 11, it is evident that the overall packet loss rates for PMQECN and P4QCN are significantly lower than those for DCQCN and TCN. This observation is attributed to the two-point architecture reacting faster to congestion, enabling prompt reduction in packet transmission rates at the sender. PMQECN exhibits a slightly higher packet loss rate compared to P4QCN, which we attribute to the higher code complexity of PMQECN relative to P4QCN, based on our analysis. Due to equipment constraints, our experiments were conducted using a software simulation platform based on P4, which cannot achieve the high-speed processing capabilities of real programmable switches. Consequently, each increase in operations on the data plane results in a certain degree of performance degradation and packet loss. Furthermore, PMQECN is a strategy for traffic QoS, aimed at ensuring QoS objectives in scenarios where multiple traffic types compete for resources.

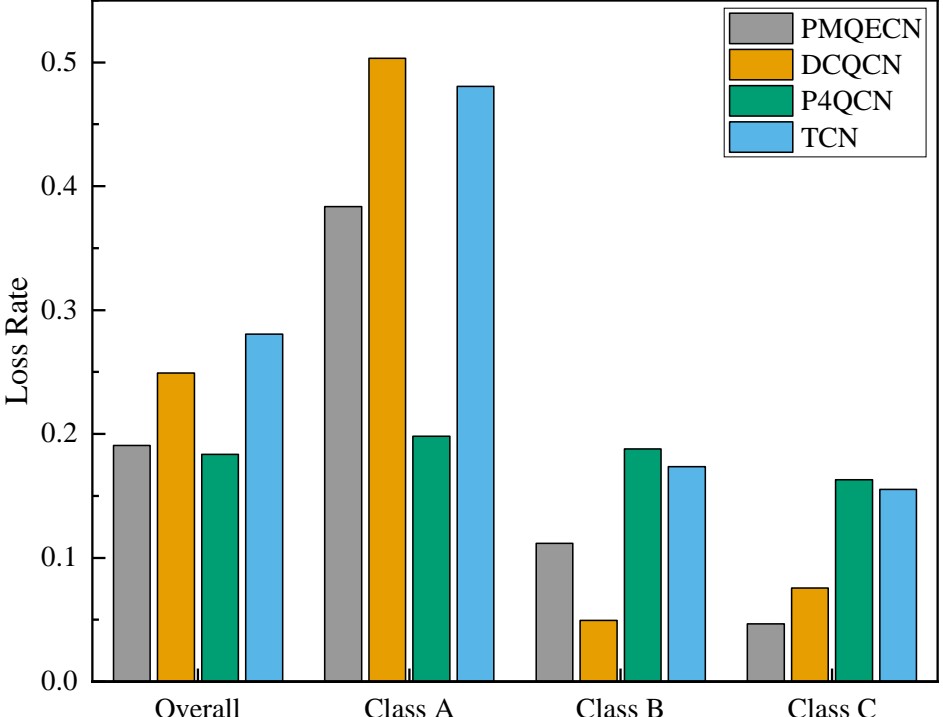

**Figure 11 Scenario 1: Packet loss rates of PMQECN, DCQCN, P4QCN, and TCN.** Where, Overall is the overall packet loss rate, and the groups of Class A, Class B, and Class C represent the packet loss rate of traffic sent from $H1$, $H2$, and $H3$ to $H5$, respectively.

According to the model established in Eq. (13). Class C's QoS objectives include the lowest acceptable packet loss rate. Therefore, compared to the minor differences in packet loss rates across different traffic types for P4QCN, PMQECN ensures the minimum packet loss rate requirement for Class C.

Compared to DCQCN, P4QCN, and TCN, PMQECN demonstrates significant improvements in queue length and RTT. In contrast to other algorithms, PMQECN reduces throughput fluctuations while maintaining throughput performance. From the packet loss rates, it is evident that PMQECN effectively maintains QoS for traffic according to specified optimization goals. In conclusion, PMQECN meets the expected performance requirements based on our evaluation.

## Scenario 2: Fairness test

**Scenario 2:** First, $H1$ sends Class A packets to $H5$, followed by $H2$ sending Class B packets to $H4$. The initial sending rate at the RP point is set to 1Mbps, and the initial phase follows an actively increasing pattern. We measured the port throughput and packet loss rate at the receiving ends $H4$ and $H5$ to observe whether congestion on one link affects another link passing through the same switch.

In Fig. 12, $q1$ represents the traffic from $H1$ to $H5$, and $q2$ represents the traffic from $H2$ to $H4$. In the event of congestion occurring in $q1$, we observe whether the other traffic

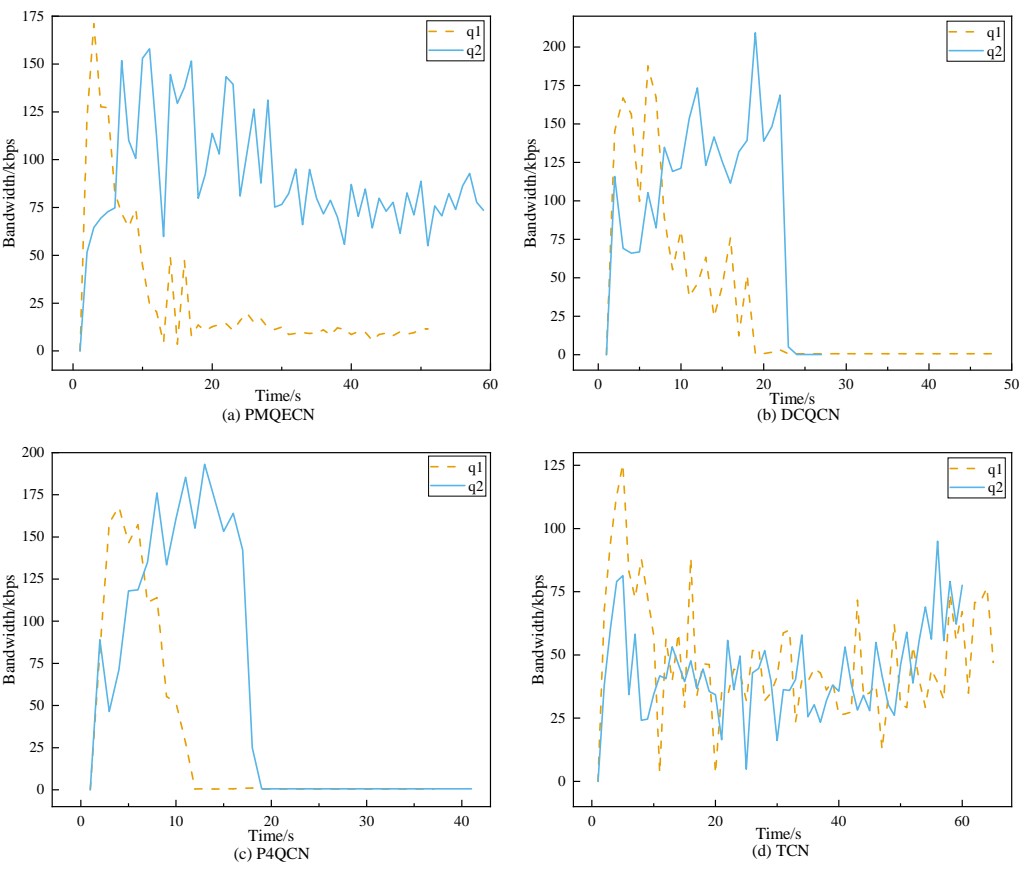

**Figure 12** **Scenario 2: Port throughput performance of q1, q2 on (A) PMQECN, (B) DCQCN, (C) P4QCN and (D) TCN.**

is affected, specifically if there are occurrences of ECN error marking. From Fig. 12, it is evident that after congestion occurred in $q1$, both DCQCN and P4QCN experienced a sharp decline in throughput for $q2$. When $q1$'s throughput reached its lowest point, $q2$'s throughput also decreased accordingly. This indicates that $H2$ received ECN feedback packets, which reduced its sending rate, thereby being influenced by the $q1$ link. Although TCN uses packet delay as the marking threshold, its actual value is adjusted based on packet delay. When congestion occurs, packets that were previously in the same queue may still interact with each other. After congestion occurred in $q1$, our algorithm ensured that $q2$'s throughput did not experience a sudden drop; instead, it exhibited a slight decrease over time but remained within a consistent range. Even when $q1$'s throughput hit its lowest point, $q2$'s throughput maintained its previous level. Our algorithm operates on the basis of queue marking, applying ECN markings tailored to different flows. This approach ensures that different flows within the same queue can operate independently without being erroneously marked due to congestion from other flows, thus preventing drastic decreases in throughput. Figure 13 illustrates the comparison of packet loss rates between $q1$ and $q2$ after passing through $S1$ and $S2$. It shows that DCQCN, P4QCN, and TCN

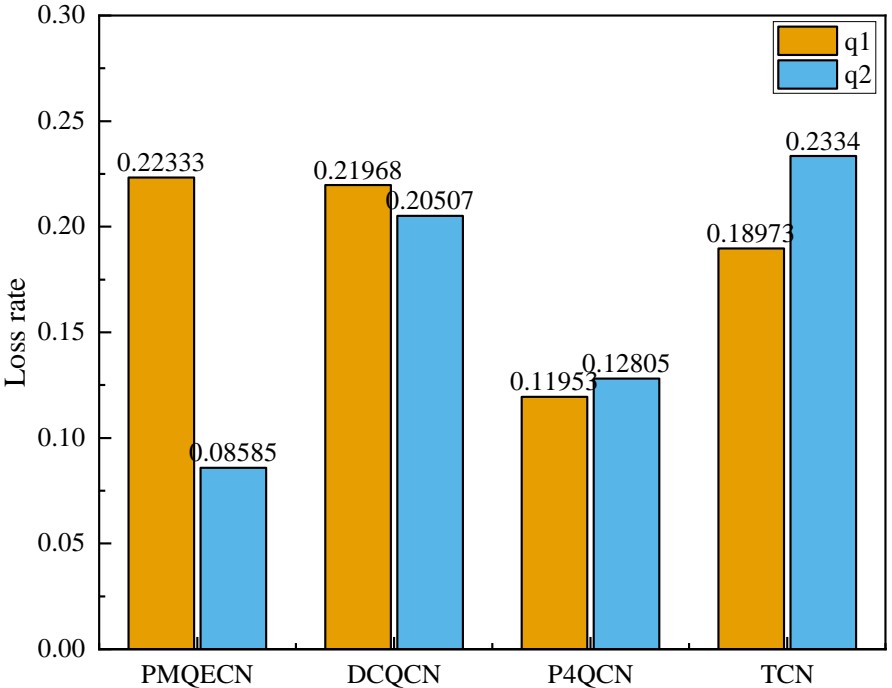

**Figure 13** Scenario 2: Packet loss rate comparison between *q1* and *q2* based on PMQECN, DCQCN, P4QCN and TCN.

exhibit packet loss rates for *q2* that are similar to or even exceed those of *q1*. In contrast, PMQECN maintains a lower packet loss rate for *q2* even when *q1* experiences higher packet loss rates due to congestion, indicating that *q2* is less affected by severe congestion in *q1* and the fairness between queues is guaranteed.

Based on the aforementioned tests, it is evident that compared to DCQCN, P4QCN, and TCN, PMQECN shows relatively minor impact of *q1* congestion on *q2*. Even when one link experiences congestion through the same switch, traffic on the other link continues to operate normally, with no significant changes in throughput and packet loss rate resulting from congestion on the other link. Therefore, we are confident that PMQECN can effectively ensure fairness between queues.

## CONCLUSION AND FUTURE WORK

In the article, we propose a multi-queue ECN marking strategy based on virtual queues and multiple QoS levels. It simulates a programmable weighted round-robin queue scheduling mechanism by using virtual queues, dividing different flows into virtual queues. It constructs a target optimization model based on the QoS of different flows and solves the model to obtain ECN marking thresholds and queue weights for each flow. The mechanism uses a two-point architecture to mitigate the issues of end-to-end latency and packet loss, and employs a dynamic threshold marking strategy to ensure stability. We validated the performance of our algorithm on the programmable testing platforms Mininet and BMv2,

and compared it against other algorithms in a simulated environment. Compared to DCQCN, P4QCN, and TCN, our algorithm reduced queue lengths within switches by 56%, 31%, and 17%, respectively. Similarly, it reduced RTT by 89.4%, 5.3%, and 47%, respectively. This strategy can maintain the QoS target of traffic as effectively as possible under the scenario of resource competition among different flows. Fairness tests indicate that our strategy effectively ensures fairness between queues. Finally, we believe that this strategy has achieved the expected outcomes.

Due to the P4 programmable software platform still being in the development stage, some features are still incomplete. Additionally, the experiment is limited by the performance of the virtual switches. In future work, we will validate the performance of this strategy in large-scale networks and optimize its mathematical model to accommodate more complex scenarios involving a wider variety of flows.

### Funding
This work was supported by Science and Technology Project of Hebei Education Department ZD2022102. The funders had no role in study design, data collection and analysis, decision to publish, or preparation of the manuscript.

### Grant Disclosures
The following grant information was disclosed by the authors:
Science and Technology Project of Hebei Education Department: ZD2022102.

### Competing Interests
The authors declare there are no competing interests.

### Author Contributions
- Yazhi Liu conceived and designed the experiments, performed the experiments, analyzed the data, performed the computation work, prepared figures and/or tables, authored or reviewed drafts of the article, and approved the final draft.
- Xinyi Yao conceived and designed the experiments, performed the experiments, analyzed the data, performed the computation work, prepared figures and/or tables, authored or reviewed drafts of the article, and approved the final draft.
- Zhigang Yang analyzed the data, performed the computation work, prepared figures and/or tables, authored or reviewed drafts of the article, and approved the final draft.
- Wei Li performed the experiments, performed the computation work, authored or reviewed drafts of the article, and approved the final draft.

### Data Availability
The code and data are available in the Supplemental File.
The code is also available at GitHub and Zenodo:
- https://github.com/YXY-1998/OUR-ECN

- YXY-1998. (2024). YXY-1998/OUR-ECN: New release-Some comments have been updated. (v.1.0.0). Zenodo. https://doi.org/10.5281/zenodo.13334643.

## Supplemental Information

Supplemental information for this article can be found online at http://dx.doi.org/10.7717/peerj-cs.2382#supplemental-information.

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
