# Peer review of "A multi-queue-based ECN marking strategy for multi-class QoS guarantee in programmable networks"

_PeerJ Computer Science, doi:10.7717/peerj-cs.2382_

## Round 0.1 · original submission · Major Revisions

Please revise the work according to the comments carefully. Then it will be evaluated again.

Reviewer 1 ·

Basic reporting

In Algorithm 1, what does packet.INT signify? I assume INT here is the abbreviation for In-band Network Telemetry. However, the full form of INT is not mentioned in the paper. It would be better to define any such parameter or header in advance.

In Table 2, the notation $i$ is mostly used to represent the traffic type; however, while describing Pm, the notation $i$ is used for representing the queue number. It would be better to use the notation $i$ to only represent either the queue number or the traffic type, but not both.

In lines 268 and 269, $L^i_s$, $L^i_q$, and $L^i_c$ are all defined as just the length of queue $i$. Whereas, in line 291, $L^i_s$ is mentioned as the average queue length. Please clearly write whether they signify instantaneous or average queue length. Similarly, in Table 1, $L^i_q$ and $L^i_c$ denote the number of packets in the queue. Does this mean the average number or the instantaneous number of packets in the queue?

In Table 1 and Algorithm 2, $N$ is used to denote the number of packets that can be accommodated in each switch. However, in line 352 and Algorithm 3, $N$ is used to denote the number of cycles for which the fast recovery phase lasts. Please use specific notations for only one purpose.

The related work discussion is generally interesting, but should please be updated to include the latest studies in this general area in order to give a comprehensive current positioning of this research with respect to related studies. For instance, please consider
Angi, et al., "Load Profiling via In-Band Flow Classification and P4 With Howdah." IEEE TNSM (2023), DOI: 10.1109/TNSM.2023.3299729;
Brouwer, et al., "Implementation of Active Queue Management Algorithms on Programmable Network Switches: A Review." Proc. 20th SC\@RUG (2022);
de Almeida, et al., "iRED: Improving the DASH QoS by dropping packets in programmable data planes." Proc. IEEE Int. Conf. on Network and Service Management (CNSM), 2022, DOI 10.23919/CNSM55787.2022.9964949;
Li, et al., "Packet rank-aware active queue management for programmable flow scheduling." Computer Networks (2023), DOI: 10.1016/j.comnet.2023.109632;
Lhamo, et al., "RED-SP-CoDel: Random early detection with static priority scheduling and controlled delay AQM in programmable data planes." Computer Commun. (2024), DOI: 10.1016/j.comcom.2023.11.026;
Zhang, et al., "BRT: Buffer Management for RDMA/TCP Mix-Flows in Datacenter Networks." IEEE TNSM (2024), DOI: 10.1109/TNSM.2024.3387984.

Experimental design

Since Mininet relies on the host machine's resources, it would be beneficial (for those who wish to replicate the experiment) if the authors could please disclose the system details on which they ran their Mininet-based experiments. For example, was the Mininet run inside a VM or directly on the device?

Validity of the findings

While the provided sample path evaluation plots are insightful, they leave the reader wondering about the corresponding steady-state performance comparisons of the proposed algorithm vs. the considered benchmarks. Could steady-state results, ideally with statistical confidence intervals, please be added. In this context, please comment on the underlying evaluation mechanisms for achieving statistically independent replications as a basis for the evaluation of the statistical confidence intervals.

In the Experiment section, for Scenario 2, the authors provide analysis for all the comparative algorithms - DCQCN, P4QCN, and TCN. However, they do not provide any direct comparison of their own algorithm for Scenario 2. It would be helpful for the reader if the authors could please highlight what improvements their algorithm brings in Scenario 2 when compared to DCQCN, P4QCN, and TCN. This could for instance be accomplished by appropriately combining the curves into one plots; or, preferably, by adding steady-state comparisons.

Reviewer 2 ·

Basic reporting

The paper combines theoretical modelling and real-world experiments to characterise the throughput, delay and bandwidth for different algorithms using Mininet and P4.
Please check that the repetitive characters as in line 52, or the absence of spaces between words have been correctly identified as in lines 70, 75, 79, 101 and so on ... Note that in some cases the acronyms do not have a space between the acronym and the meaning, as in lines 142 ... 431 .... There are also missing spaces between the citations in lines 446, 451 ...
Please make the following changes to the figures:
1. Improve the caption and the labels of the figures. Please provide more detailed and descriptive improvements to the figure captions, especially for Figure 5.
2. Add dots to all captions.
Please note that the code has been commented in Chinese and should be changed to English. Review algorithm 1 sintaxis.
It may be beneficial to structure the paper in a more natural way with the following sections:
1. Introduction
2. Related work
Missing reference in line 369 when suggesting the parameter value as in the literature.

In the Experiments section, it would be better to have two subsections for each scenario and a summary of the figures.
The legend referring to the proposed solution as "OUR" should be changed to a more appropriate designation.

Please refer to the repository as a footnote by means of using a url.

Experimental design

This work is consistent with the objectives and scope of this publication. However, it would be beneficial to consider the following minor changes:

The positioning of the paper, in terms of explaining in more detail what the current gap/problem is that this paper is trying to solve, could be improved. (At the end of the related work as a comparison to other work).

The experimental design section is clearly explained according to the principles of queueing theory, with each step presented in a logical sequence.

In addition, the design explanation should include a reference to the relevant literature, in particular the value of N = 5 in line 352.

Please make the presentation of the results figure more professional and better explanation of the algorithms.

It would be beneficial to provide further details regarding the configurations and setups in order to facilitate replication and enhance comprehension.

Validity of the findings

This paper is interesting because it compares their solution with other commonly used QCN solutions.
However, in order to improve the presentation of the result figures and the visibility of Figure 8, we recommend the use of zoom or subplots.
As previously stated, the throughput change of TCN is relatively stable but generally lower than that of the other methods. I believe it would be beneficial to include the TCN solution in Figure 10.
Please include the parameter represented in Figure 11, rather than just the units.
It would be beneficial to include a table indicating the relevant parameters of the configuration setup.
It would also be useful to represent metrics such as packet loss, as mentioned.
Finally, an overview of the results obtained would be a valuable addition to the conclusions.

---

## Round 0.2 · accepted · Accept

Thanks to the authors for their efforts to improve the article. This version successfully satisfied the reviewers. However, please pay attention to the comments. Some edits are needed at the proof stage.

Reviewer 1 ·

Basic reporting

Thank you for implementing the necessary changes. There is one more minor comment regarding the updates.
The term 'INT' should be defined by its full form (In-band Network Telemetry) when it is first used in Section 2 Related Work on Page 3, line 128, instead of on Page 6, line 267.

The abbreviation 'CV' for the coefficient of variation is introduced in line 509. However, the full term is still used in lines 511 and 516, while 'CV' appears in line 514 and Table 4. For consistency, use 'CV' after it is defined in line 509.

Experimental design

No comment

Validity of the findings

No comment

Reviewer 2 ·

Basic reporting

The authors have implemented the recommended changes in accordance with the provided guidance. This includes incorporating sitaxis and grammatical alterations, providing enhanced explanations, and restructuring the paper for optimal clarity and coherence.

Experimental design

The authors have addressed the suggestions and implemented the requested changes.
They have also provided clarification on the research gap, presented a solution with a
theoretical framework, included missing details, created more suitable figures with a clear interpretation, and provided detailed information about the scenario setup to ensure reproducibility.

Validity of the findings

The authors put forward a QCN solution that addresses the shortcomings of legacy solutions. These combine queuing theory to achieve optimal activation of ECN mechanisms. Furthermore, the authors have incorporated figures for comparison with established benchmarks, plotting their performance.
They have also included an overview of the results in their conclusions.